# Selection of *pfcrt* K76 and *pfmdr1* N86 Coding Alleles after Uncomplicated Malaria Treatment by Artemether-Lumefantrine in Mali

**DOI:** 10.3390/ijms22116057

**Published:** 2021-06-03

**Authors:** Hamma Maiga, Anastasia Grivoyannis, Issaka Sagara, Karim Traore, Oumar B. Traore, Youssouf Tolo, Aliou Traore, Amadou Bamadio, Zoumana I. Traore, Kassim Sanogo, Ogobara K. Doumbo, Christopher V. Plowe, Abdoulaye A. Djimde

**Affiliations:** 1Institut National de Sante Publique, INSP, Bamako P.O. Box 1771, Mali; hmaiga@icermali.org; 2Malaria Research & Training Center, Department of Epidemiology of Parasitic Diseases, Faculty of Pharmacy, Faculty of Medicine and Dentistry, University of Sciences, Techniques and Technologies of Bamako, Bamako P.O. Box 1805, Mali; isagara@icermali.org (I.S.); karim@icermali.org (K.T.); bila@icermali.org (O.B.T.); ytolo@icermali.org (Y.T.); aliout@icermali.org (A.T.); bamadio@icermali.org (A.B.); zouissac@gmail.com (Z.I.T.); sanogok@icermali.org (K.S.); okd@icermali.org (O.K.D.); 3Weill Cornell Medical College, Weill Cornell Medicine, New York, NY 10021, USA; grivoyannis@jhmi.edu; 4School of Medicine, University of Maryland, Baltimore, MD 21201, USA; plowe.chris@gmail.com

**Keywords:** *Plasmodium falciparum*, *Pfcrt*, *Pfmdr1*, Artemether-lumefantrine, Mali

## Abstract

Background: Artemether-lumefantrine is a highly effective artemisinin-based combination therapy that was adopted in Mali as first-line treatment for uncomplicated *Plasmodium falciparum* malaria. This study was designed to measure the efficacy of artemether-lumefantrine and to assess the selection of the *P. falciparum chloroquine resistance transporter* (*pfcrt*) and *P. falciparum multi-drug resistance 1* (*pfmdr1*) genotypes that have been associated with drug resistance. Methods: A 28-day follow-up efficacy trial of artemether-lumefantrine was conducted in patients aged 6 months and older suffering from uncomplicated falciparum malaria in four different Malian areas during the 2009 malaria transmission season. The polymorphic genetic markers MSP2, MSP1, and Ca1 were used to distinguish between recrudescence and reinfection. Reinfection and recrudescence were then grouped as recurrent infections and analyzed together by PCR-restriction fragment length polymorphism (RFLP) to identify candidate markers for artemether-lumefantrine tolerance in the *P. falciparum chloroquine resistance transporter* (*pfcrt*) gene and the *P. falciparum multi-drug resistance 1* (*pfmdr1*) gene. Results: Clinical outcomes in 326 patients (96.7%) were analyzed and the 28-day uncorrected adequate clinical and parasitological response (ACPR) rate was 73.9%. The total PCR-corrected 28-day ACPR was 97.2%. The *pfcrt* 76T and *pfmdr1* 86Y population prevalence decreased from 49.3% and 11.0% at baseline (*n* = 337) to 38.8% and 0% in patients with recurrent infection (*n* = 85); *p* = 0.001), respectively. Conclusion: Parasite populations exposed to artemether-lumefantrine in this study were selected toward chloroquine-sensitivity and showed a promising trend that may warrant future targeted reintroduction of chloroquine or/and amodiaquine.

## 1. Introduction

Resistance of *Plasmodium falciparum* to chloroquine is present in almost all malaria-endemic countries [1]. Resistance to most other antimalarial drugs emerged after these therapies were introduced to replace chloroquine [2,3,4]. Artemisinins are structurally distinct from all other antimalarials and have so far been effective against multidrug-resistant strains of *P. falciparum* [5]. The World Health Organization (WHO) recommends artemisinin-based combination therapies (ACTs) as first-line treatment for uncomplicated malaria worldwide [6]. In Mali, the National Malaria Control Program (NMCP) recommends treating uncomplicated malaria with artesunate-amodiaquine or artemether-lumefantrine [7]. Although treatment policy in Mali changed from chloroquine to these ACTs in 2006, artemether-lumefantrine was not widely implemented as first-line therapy until 2007 [7]. Studies in Mali in 2005 revealed high efficacy of artemether-lumefantrine [8,9,10].

Artemether-lumefantrine combines the rapid-acting, short half-life synthetic artemisinin with the long-acting, more slowly cleared aryl amino alcohol, lumefantrine [11]. Efforts have been made in past years to identify mechanisms of decreased sensitivity to antimalarial drugs by searching for the selection of particular genetic polymorphisms in parasites after treatment. We previously showed results of treatment with chloroquine, pyrimethamine, pyrimethamine-sulfadoxine, artesunate-amodiaquine, and artesunate-sulfadoxine-pyrimethamine that were respectively selected for parasites carrying the molecular markers of resistance to each drug [12,13,14,15].

The *Plasmodium falciparum multi-drug resistance 1* (*pfmdr1*) gene on chromosome 5 encodes a digestive vacuole transmembrane glycoprotein (Pgh1, for P-glycoprotein homologue 1). Point mutations at codons 86, 184 (predominant in Asia and Africa) and 1034, 1042, and 1246 (predominant in South America) and increased gene copy numbers of *pfmdr1* are associated with resistance to many drugs, including chloroquine, quinine, mefloquine, and the aryl amino alcohols (halofantrine and lumefantrine) [16,17,18].

Mutations in the *P. falciparum chloroquine resistance transporter* (*pfcrt*) gene on chromosome 7, which also encodes a digestive vacuole transmembrane protein, are associated with chloroquine resistance [19]. Point mutations at codon 76 may play a role in mefloquine resistance and are now suggested to contribute to parasite susceptibility to other aryl amino alcohols, particularly lumefantrine [20].

Several countries in West Africa, including Mali, use artemether-lumefantrine as a first-line treatment for uncomplicated malaria. A number of other studies from Africa have investigated whether artemether-lumefantrine administration selects for particular *pfcrt* and *pfmdr1* SNPs, but just a few were from West Africa [21] and Mali. Here, a clinical study was performed at four different Malian field sites, with different malaria transmission patterns and different drug-resistance rates, to monitor the efficacy of artemether-lumefantrine and its potential impact on *pfmdr1* and *pfcrt* polymorphisms three years after the wide use of artemether-lumefantrine in Mali.

## 2. Results

### 2.1. Trial Profile

Of 400 recruited patients, 337 were enrolled: 77 in Kolle (*n* = 77), 88 in Faladje (*n* = 88), 100 in Bandiagara (*n* = 100), and 72 in Pongonon (*n* = 72). After 28 days of follow-up, 327 (97%) participants completed the study: 70 in Kolle (91%), 88 in Faladje (100%), 99 in Bandiagara (99%), and 70 in Pongonon (97.2%), as shown in Figure 1.

### 2.2. Characteristics of Participants at Inclusion

Table 1 shows some differences exist among the four study sites, notably gametocyte carriage and parasite density at enrollment. Average parasite density at enrollment was 37,250 and average gametocyte carriage was 6.5% to 25% at one study site.

### 2.3. Treatment Efficacy for Uncomplicated Falciparum Malaria

Table 2 shows the 28-day treatment outcomes. Only one early treatment failure was observed: a child developed convulsions on Day 1 with a persistent fever (38.6 °C) despite diminished parasitemia (45,275/μL to 100/μL), and no death was observed in this study. The earliest recurrent infections were detected on Day 14. Between Days 14 and 28, 86 cases of parasitemia occurred, resulting in a cumulative risk of failure of 26.1%. Genotyping using the three polymorphic genes (MSP2, MSP1, and Ca1) showed that nearly all recurrent infections were new infections, rather than recrudescence. Considering only recrudescence, eight (2.5%) participants were classified as having genuine treatment failures, and risk of recrudescence did not differ significantly among the four sites (*p* > 0.78).

### 2.4. Secondary Outcomes

The clearance of fever was not statistically different among the four sites (data not shown). By Day 2, nearly all fevers had resolved. Parasitemia was statistically different among sites and highest in Pongonon on Day 1, but parasites were undetectable in all patients by Day 3 (Figure 2). Hemoglobin concentrations improved equally in all four sites (data not shown). Gametocytemia was highest in Pongonon at enrollment, but the levels became statistically similar to other sites by Day 1 (Figure 3). Gametocytes were not detected on any peripheral blood smears from Bandiagara.

### 2.5. Baseline Allele-Prevalence and Treatment Outcome

A total of 337 infections before treatment were successfully analyzed. Figure 4 and Figure 5 show the prevalence of the studied *pfcrt* 76T and *pfmdr1* 86Y mutations in all participants at baseline (Day 0) and re-treatment (Day R). Carriage of the *pfmdr1* N86 allele and/or the *pfcrt* K76 allele (wild types) was not associated with blood smear positivity after initial parasite clearance in participants at any site within the 28-day follow-up period (*p* > 0.05). Additionally, we conducted an analysis of directional selection acting on alleles *pfmdr1* codon 86 and *pfcrt* codon 76 from infected patients following therapy with artemether-lumefantrine. The frequencies of the *pfmdr1* N86 and the *pfcrt* K76 alleles in recurrent infections increased from 84.7% to 100% (*p* = 0.0002) and 43.5% to 61.2% (*p* = 0.02), respectively (Table 3). The direction of the allele changes from baseline to recurrence was assessed using McNemar’s χ^2^ test for paired samples.

Examples of gel photo illustrations for MSP2, MSP1, and Ca1 (Figure 6).

## 3. Discussion

We found that both *pfcrt* K76 and *pfmdr1* N86 (wild types) were selected in vivo after treatment with artemether-lumefantrine. Statistically significant *pfcrt* allele-selection has not been observed as often as *pfmdr1* allele-selection. Our study, in addition to those by Sisowath, et. al. [22], and Otienoburu et al. [23], confirmed that the selection of the K76 SNP occurs in response after artemether-lumefantrine treatment. Veiga et al. showed the selection of K76 and N86 with lumefantrine treatment using gene-edited parasites [24]. All these studies described a decreasing chloroquine exposure and increased lumefantrine exposure and that this contributed to the selection of the wild types of *pfmdr1* and *pfcrt*.

Whether *pfcrt* K76 and *pfmdr1* N86 are selected for under the same mechanism remains to be determined. The reason for recurrent infections is primarily reinfection and allele selection occurs in reinfecting parasites. The concentration of lumefantrine is high enough to eliminate mutated parasites, but insufficient to eliminate wild-type parasites without the combined protection of artemether. Hastings and Ward have suggested that the selection of N86 may represent a marker of tolerance to lumefantrine [25].

In our study, treatment with artemether-lumefantrine (AL) was selected for *pfcrt* K76 and *pfmdr1* N86 alleles (wild types) among recurrent infections. In this analysis, allele prevalence on Day 0 was compared with allele prevalence on Day R among the 85 patients who had recurrent infection. Recurrent parasitemia was statistically similar among three of the four sites (Kolle, Faladje, and Bandiagara), though recurrent malaria was most commonly observed in Bandiagara (15.2%). Recurrent malaria (4.3%) and parasitemia (2.9%) were both statistically lower in Pongonon.

Carriage of the *pfmdr1* N86 allele has been associated with treatment failure in patients given artemether-lumefantrine [26,27,28,29,30,31,32,33,34,35,36,37,38,39,40,41,42,43,44,45,46,47]. Prevalence of the *pfmdr1* N86 allele also increased in all isolates, which shows a selection in favor of the wild-type allele, as seen previously with the use of artemether-lumefantrine [48]. The pattern of the *pfmdr1* allele prevalence change was similar to that reported recently by Dokomajilar [49] from their study of Ugandan children, from Humphreys in Tanzania [26], and from Sisowath in Unguja Island and Micheweni [20]. Prevalence of the *pfcrt* K76 allele (wild type) increased in all isolates after treatment. We found evidence for significant selection of both *pfcrt* K76 and *pfmdr1* N86 SNPs (wild types) after treatment with artemether-lumefantrine.

The selection of a SNP previously associated with quinoline antimalarial drug resistance was first documented in 2005. Our results regarding *pfmdr1* N86 allele-selection (wild type) are consistent with previous studies; however, the risk of new infections among these studies has varied. In areas of Tanzania with intermediate transmission, the risk of new infection after artemether-lumefantrine treatment was only 5%; a similar study performed in an area of high transmission in Tanzania showed new infection rates of about 50%. In our study, risk of reinfection also varied by site, from Pongonon to Faladje/Bandiagara/Kolle. As transmission intensity increased from one site to another, so did the proportion of treated patients selecting for parasites harboring a particular allele. In vitro studies propose that gene selection occurs during the elimination phase of lumefantrine (t ½ = 3–6 days) [50], but this effect is unclear at the population level, where the proportion of parasites exposed to drug-selective pressures will vary geographically.

The eight (2.5%) recrudescent infections in this study could not be explained by *pfmdr1* or *pfcrt* selection alone, albeit the sample size may have been insufficient to identify any associations. We thus relied on the combined outcomes of reinfection and recrudescence, as previous studies have done. However, grouping these outcomes involves the risk of seeing relationships that may not exist. Unless an increased frequency of recrudescence occurs in patients treated with artemether-lumefantrine, insights regarding tolerance and resistance will remain limited.

The presence of these wild-type alleles has been associated with reduced sensitivity to lumefantrine in vitro [51]. Other studies have associated the *pfmdr1* haplotype of N86-184F-1246 with treatment failure [52,53]. We hypothesize that treatment with artesunate-amodiaquine (ASAQ) may lead to a selection of molecular markers of drug resistance (*pfcrt* 76T and *pfmdr1* 86Y (mutant types)) against artemisinin partner molecules [54,55]. AL and ASAQ could have opposite effects in selecting for *pfcrt* K76 and *pfmdr1* N86 alleles (wild types), suggesting a potential benefit of using different ACTs simultaneously as first-line treatment to reduce selective pressures by each regimen. Alternating the use of AL versus ASAQ could maintain drug effectiveness, while decreasing the risk of tolerance/resistance. Larger studies examining this hypothesis could inform NMCPs to adjust malaria treatment policies.

In conclusion, parasite populations exposed to artemether-lumefantrine in this study were selected toward chloroquine-sensitivity and showed a promising trend that may warrant future targeted reintroduction of chloroquine or/and amodiaquine in areas where *P. falciparum* was previously resistant to these drugs.

## 4. Methods

### 4.1. Study Area and Population

This study was conducted during the malaria transmission season, from October 2009 to January 2010 in four different Malian areas: Kolle, Faladje, Bandiagara, and Pongonon. Kolle is a rural village of ~3000 inhabitants located 57 km southwest of Bamako. The climate is Sudanian savanna with two distinct seasons, the dry season (January–June) and the rainy season (July–December). Malaria is hyperendemic during 3 to 4 months of the rainy season with a parasite index (PI) of 70%–85%, and of 40%–50% during the dry season [56]. The entomological inoculation rate during the high transmission season was 5.2 infective bites per person per month and the *pfcrt* K76T mutation prevalence was previously recorded in 1999 and 2002 at 28.3% and 85%, respectively [57,58]. Faladje is located 80 km northwest of Bamako and its health center serves ~23,000 inhabitants. Malaria is seasonal (July–November) with a hyperendemic peak in October. The *pfcrt* K76T mutation prevalence was previously recorded in 2003 at 80% [57]. Bandiagara is a town of ~13,364 inhabitants located 700 km northeast of Bamako. Malaria is endemic with a transmission period from July to November. The *pfcrt* K76T mutation prevalence was previously recorded in 2002 at 39% [57]. Pongonon is a village of ~1400 inhabitants located 20 km from the rural town of Koro and 785 km from Bamako in northeastern Mali. Malaria is endemic with a transmission period from July to October. The entomological inoculation rate was one infective bite per person per month. Pongonon is located in an ecological transition zone between the Savanna and Sahel regions. The *pfcrt* K76T mutation prevalence was previously recorded in 2002 at 13% (unpublished data).

### 4.2. Study Volunteers

Inclusion criteria consisted of the following: (i) age ≥6 months, (ii) microscopic detection of *P. falciparum* at a density ≥2000 asexual parasites/µL to 200,000/μL of blood, (iii) axillary temperature ≥37.5 °C or reported fever in the preceding 24 h, (iv) ability to swallow oral medication, and (v) willingness to consent and comply with the study protocol for the duration of the study (parental consent was required for children <18 years). Exclusion criteria consisted of the following: (i) signs of severe malaria according to 2003 WHO definitions, (ii) a concomitant febrile condition (i.e., measles, acute respiratory infection, severe diarrhea), (iii) severe malnutrition, (iv) disclosed or clinically patent pregnancy, (v) other known underlying chronic or severe diseases, or (vi) history of hypersensitivity to artemether-lumefantrine or the rescue treatment, artesunate-amodiaquine.

### 4.3. Sample Collection

Finger-prick blood samples were collected from enrolled patients to assess the parasite density and hemoglobin level on-site, and to be preserved on the filter papers (Whatman 3MM, Whatman Lab Sales Ltd., Maidstone, Kent, UK), which were stored in self-sealing plastic bags with desiccant for later analysis. Patients received a twice-daily dose of artemether-lumefantrine of 20 mg/120 mg as indicated by weight for three consecutive days: one tablet per dose for patients weighing 5–14 kg, two tablets per dose for those weighing 15–24 kg, three tablets per dose for those weighing 25–34 kg, and four tablets per dose for those weighing ≥35 kg. All doses were administered on-site under supervision. A full dose was re-administered if a participant vomited within 30 min of the initial drug administration. After treatment initiation, patients were examined and finger-prick blood samples were collected routinely on Days 1, 2, 3, 7, 14, 21 and 28 of the treatment and on any day of recurrent illness during the 28-day follow-up period. In cases of treatment failure, quinine was administered if the patient was re-diagnosed with uncomplicated malaria or parasitemia. As per the national guidelines, patients were referred to a hospital and intravenous quinine was administered if severe malaria was diagnosed at any time during the follow-up period. Treatment was also given for any other conditions diagnosed (Appendix A).

### 4.4. Laboratory

Specimen analysis was completed at the Malaria Research and Training Center (MRTC) lab in Bamako, Mali. Parasite density and rate of gametocyte carriage were measured, as number of parasites per microliter, from Giemsa-stained thick blood smears. A positive smear was defined as a thick smear with at least one asexual parasite per 100 fields under 1000× magnification (standardized technique from the Malian Ministry of Health). Blood was collected for hemoglobin measurement (HemoCue AB) on Day 0 prior to treatment and on Days 14 and 28 of treatment. Children with hemoglobin <7 g/dL were treated with oral iron therapy according to national policy [59]. For quality control purposes, 10% of smears were read by two independent microscopists blinded from the results of the first reading. On the basis of the results of these assessments, patients were classified as having therapeutic failure (early or late) or an adequate clinical and therapeutic response.

### 4.5. DNA Extraction

Three methods were used for DNA extraction. (1) Methanol was used to extract parasite DNA from all filter paper blood samples. Samples that did not yield DNA by this method were extracted using (2) the Chelex method [60]. For a single day 0 sample from Bandiagara, a filter paper blood sample was not available; thus, parasites were extracted from the thick smear according to a modified dried blood protocol using (3) QIAmp DNA blood mini kit (Qiagen, Valencia, CA, USA). DNA samples were stored at −20 °C until use.

### 4.6. Genotyping of Plasmodium Falciparum Isolates

MSP2, MSP1, and Ca1 were used to compare genotypic profiles between pre- and post-treatment *P. falciparum* isolates, in order to distinguish between episodes of recrudescence and reinfection [61,62,63]. The number of clones in pre- and post-treatment *P. falciparum* isolates were denoted by the number of different-sized PCR products [61]. Length differences were determined using a 100 bp molecular weight ladder (Promega, Madison, WI, USA). Allele-specific positive controls and DNA-free negative controls were included in each round of reactions. Primers for MSP1, MSP2 and Ca1 were synthesized by Integrated DNA Technologies, Coralville, IA, USA.

### 4.7. Pfcrt Codon 76-Point and Pfmdr1 Codon 86-Point Mutations Analysis

Mutations in *pfcrt* and *pfmdr1* were assessed for all baseline infections (Day 0) and all recurring infections post-treatment (Day R) to examine any association between the artemether-lumefantrine treatment and the development of tolerance/resistance, as has been previously suggested [21]. The *pfcrt* K76T SNP was analyzed by restriction fragment length polymorphism (RFLP)-PCR and mutation-specific MS-PCR [13]. DNA from *P. falciparum* isolates Dd2 (containing the 76T mutation) and 3D7 (lacking this mutation) were used as the positive controls, while a DNA-free reaction mix was used as negative control. The *pfmdr1* N86Y SNP was analyzed by PCR-RFLP, as described previously [64]. The negative control without parasite DNA and the positive controls were processed alongside the samples. HB3 and Dd2 *P. falciparum* isolates, representing wild-type (N86) and mutant genotypes (86Y), respectively, were used as the positive controls. PCR was performed in a tetrad 2 thermocycler (BIO-RAD, Marnes-la-Coquette, France). Primers for *pfcrt* were synthesized by Integrated DNA Technologies, Coralville, IA, USA. Primers for *pfmdr1* were synthesized by the UMD Biopolymer core facility. RFLP and MS-PCR products and digestions were separated by electrophoresis in 2% agarose gels with ethidium bromide and were visualized under UV transillumination (SynGene; GeneSnap v.6.05).

### 4.8. Data Management and Analysis

Data were collected on case report forms and double entered using MS Access and analyzed using Stata (Stata Corp 11). The prevalence of single mutations and of various genotypes including the double mutant (*pfcrt* 76T + *pfmdr1* 86Y) were calculated with 95% confidence interval (CI). Chi-square or Fisher exact probability tests and McNemar’s χ^2^ test were used for comparisons as appropriate with statistical significance set at *p* value < 0.05.

### 4.9. Ethics

The study was approved by the ethical committee of the Faculty of Medicine, Pharmacy and Dentistry of Bamako, University of Sciences, Techniques, and Technologies of Bamako, Mali #N^o^9/63/FMPOS#. Community permission was obtained from each locality prior to the study. Written informed consent was obtained from all adults and parents/guardians of children enrolled prior to screening.

## Figures and Tables

**Figure 1 ijms-22-06057-f001:**
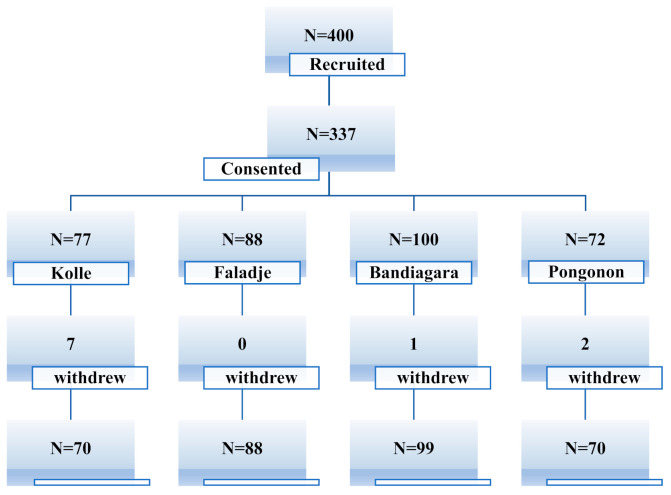
Study profile.

**Figure 2 ijms-22-06057-f002:**
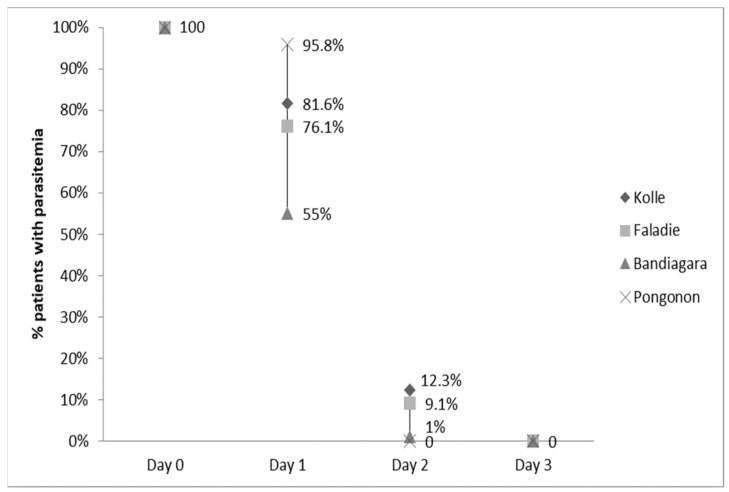
Parasitemia clearance.

**Figure 3 ijms-22-06057-f003:**
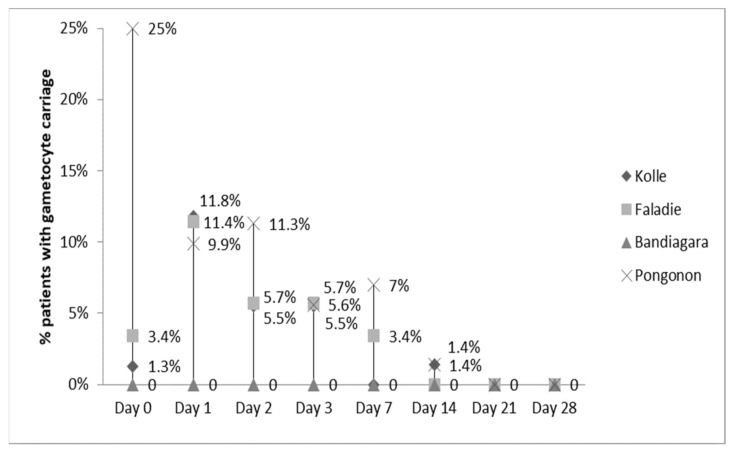
Gametocytemia clearance.

**Figure 4 ijms-22-06057-f004:**
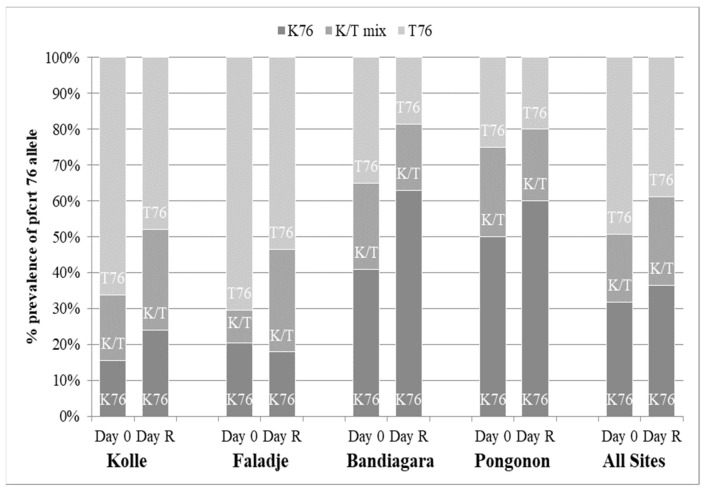
*Pfcrt* allele frequency. D0 = date of enrollment; Day R = date of recurrent infection.

**Figure 5 ijms-22-06057-f005:**
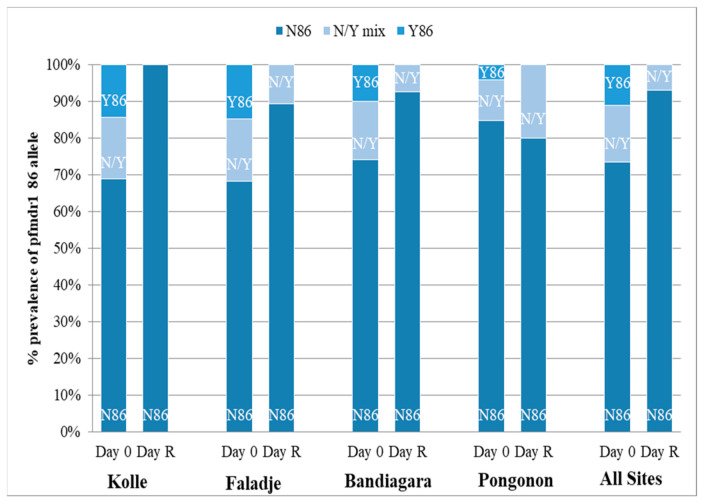
*Pfmdr1* allele frequency. D0 = date of enrollment; Day R = date of recurrent infection.

**Figure 6 ijms-22-06057-f006:**
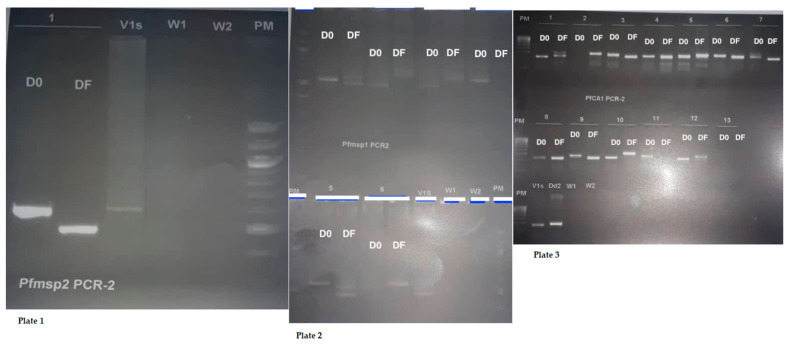
Gel photo illustrations for MSP2, MSP1, and Ca1 (D0 = Day 0 and DF = Failure Day). (**Plate 1**). DNA band of *Plasmodium falciparum* MSP2. (**Plate 2**). DNA band of *P. falciparum* MSP1. (**Plate 3**). DNA band of microsatellite Ca1.

**Table 1 ijms-22-06057-t001:** Sociodemographic, clinical, and parasitological variables at enrollment at each study site.

	Kolle	Faladje	Bandiagara	Pongonon
Female (%)	42 (54.6%)	41 (46.6%)	49 (49.0%)	41 (56.9%)
Age in years (% under 5) *	6 (4–7) (41.6%)	6 (4–7) (35.2%)	7 (5–11) (20.0%)	5 (4–7) (36.1%)
Axillary Temperature (°C) *	38.4 (37.3–38.9)	38.4 (37.6–39.2)	38.2 (37.0–39.2)	38.2 (27.6–39.0)
Parasite density (per uL) *	48,325 (27,075–71,475)	48,075 (30,525–86,975)	38,000 (16,200–80,250)	23,813 (12,363–35,425)
Gametocyte carriage (%)	1 (1.3%)	3 (3.4%)	0 (0.0%)	18 (25.0%)
Hemoglobin (mg/dL) ^†^	10.0 (1.8)	10.8 (1.9)	11.2 (2.1)	10.3 (1.9)
Antimalarial used in previous 2 weeks (%)	1 (1.3%)	1 (1.1%)	9 (9.0%)	0 (0.0%)

Data are number (%), unless otherwise indicated. * Median (IQR) ^†^ Mean (SD). mg—milligram; dL—deciliter; uL—microliter.

**Table 2 ijms-22-06057-t002:** Primary treatment outcomes.

	Kolle N = 70 n (%)	Faladje N = 88 n (%)	Bandiagara N = 99 n (%)	Pongonon N = 70 n (%)	*p*-Value
Early treatment failure (ETF)	1 (1.4)	0 (0.0)	0 (0.0)	0 (0.0)	-
Late clinical failure (LCF)	9 (13.0)	5 (5.7)	15 (15.2)	3 (4.3)	0.044
Late parasitological failure (LPF)	17 (24.6)	22 (25.0)	12 (12.1)	2 (2.8)	0.0001
Crude Adequate clinical and parasitological response (ACPR)	43 (62.3)	61 (69.3)	72 (72.7)	65 (92.8)	0.0001
Reinfection ^†^	25 (35.7)	24 (27.3)	23 (23.2)	4 (5.7)	0.0001
ACPRc ^‡^	68 (97.1)	85 (96.6)	95 (96.1)	69 (98.6)	0.85

^†^ Reinfection. ^‡^ Adequate clinical and parasitological response correction (ACPRc) by PCR.

**Table 3 ijms-22-06057-t003:** Matched post-treatment allele frequencies.

	Prevalence (%) in Patient Samples (No. of Patients With Allele/Total No. in Group)
Baseline	Recurrence	χ^2^	*p-*Value	Selection
*Pfcrt* K76T					
K	43.5 (37/85)	61.2 (52/85)	6.08	0.02	K allele
T	56.5 (48/85)	38.8 (33/85)			
*Pfmdr1* N86Y					
N	84.7 (72/85)	100 (85/85)	13.00	0.0002	N allele
Y	15.3 (13/85)	0 (0/85)

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
