# Peer review of "Selection of *pfcrt* K76 and *pfmdr1* N86 Coding Alleles after Uncomplicated Malaria Treatment by Artemether-Lumefantrine in Mali"

_ijms, 2021, doi:10.3390/ijms22116057_

Round 1

Reviewer 1 Report

 Overall I am happy with the presentation and the writing of the manuscript. I have a few concerns mentioned below.

Results 3.3 – I would like the authors to show gel image of a few samples (as supplement) for the three polymorphic genes (MSP2, MSP1, and Ca1) that distinguishes between recurrent and new infections, considering that the majority of the population was N86 in Figure 3 at Day=0. Previously, gene edited data has suggested strong selection of the N86 allele with LUM treatment.

Discussion section – Missing key citations that show selection of K76 and N86 with LUM treatment using gene-edited parasites (eg. Veiga et al. Nat Commun)

“Our findings suggest pfcrt K76T and pfmdr1 N86Y mutations are associated with enhanced Plasmodium falciparum susceptibility to artemether-lumefantrine, and implicate these alleles for the first time as potential markers of artemether-lumefantrine tolerance in West Africa.”

This statement needs to be removed as multiple previous studies have shown this before included studies using gene edited parasites.

Author Response

Comments and Suggestions for Authors - 1

 Overall, I am happy with the presentation and the writing of the manuscript. I have a few concerns mentioned below.

Response: Thank you

Results 3.3 – I would like the authors to show gel image of a few samples (as supplement) for the three polymorphic genes (MSP2, MSP1, and Ca1) that distinguishes between recurrent and new infections, considering that the majority of the population was N86 in Figure 3 at Day=0. Previously, gene edited data has suggested strong selection of the N86 allele with LUM treatment.

Response: Gel image is added. Thank you

Discussion section – Missing key citations that show selection of K76 and N86 with LUM treatment using gene-edited parasites (eg. Veiga et al. Nat Commun)

Response: Reference is added. Thank you

« M. Isabel Veiga, Satish K. Dhingra, Philipp P. Henrich, Judith Straimer, Nina Gnädig, Anne-Catrin Uhlemann, Rowena E. Martin, Adele M. Lehane & David A. Fidock. Globally prevalent PfMDR1 mutations modulatePlasmodium falciparum susceptibility to artemisinin-based combination therapies, Nature Communications volume 7, Article number : 11553 (2016)”

“Our findings suggest pfcrt K76T and pfmdr1 N86Y mutations are associated with enhanced Plasmodium falciparum susceptibility to artemether-lumefantrine and implicate these alleles for the first time as potential markers of artemether-lumefantrine tolerance in West Africa.”

This statement needs to be removed as multiple previous studies have shown this before included studies using gene edited parasites.

Response: Sentence is removed, Thank you

Reviewer 2 Report

The authors describe here a study performed with 326 patients with uncomplicated falciparum malaria in Mali that measures the effect of antimalarial drug combination- artemether-lumefantrine, in terms of two genotypes - pfcrt and pfmdr1 genes that have been associated with drug resistance. The authors show that recurrent infections occur in 26% of the patients and these recurrent parasite population showed a significant shift towards the pfcrt K76 and pfmdr1 N86, pointing to association of these genotypes with AL drug treatment failure. The study is performed well and the results presented show statistical rigour. However, the authors do not show any novel findings and their results just confirm results from similar clinical studies in Africa, now extending it to Mali. This should not however exclude the work from publication.

Major comments-
1) Line 238- I could not understand this sentence. It would help if the authors could rephrase this in more simpler or layman terms.
2) Line 260 - At several instances like in this para, the authors talk about (resistant or susceptible), (mutant or wild), (parasites or alleles). It would make the discussion more readable and understandable if they could add the genotype notation (K76 or T76 etc) in brackets next to their assertions. I had to keep going back to their results or introduction to understand which allele is resistant/susceptible to which drug.
3) Line 261, Do you mean that allele selection occurs in reinfecting parasites rather than recrudescent parasites? Do you have enough data to support this assertion? I dont see any comparison of prevalence of mutations in reinfecting vs recrudescent. Given that there is only 8 cases of recrudescence in the dataset, probably we cant even make that comparison with reasonable statistical confidence. Maybe just say reason for recurrent infections is primarily reinfection and allele selection occurs in reinfecting parasites. 
4) Line 278 - You can be more specific as to how the pattern of pfmdr1 allele change is similar among all the studies.
5) Line 284 - This paragraph need to be revised. I could not understand the point the authors are trying to make. 

Minor comments-
1) Line 50 - remove apostrophe in Artemisinin.
2) Line 78 - I dont know if "however" is appropriate here.
3) Line 81 - "Here", a clinical study was performed...
4) Line 82 - "pattern" to "patterns"
5) Line 214 - "any death" to "no death"?
6) Line 220 - missed bracket
7) Line 305 - "hypothesized" to "hypothesize"
8) Line 63 -  There are a ton of studies that have showed drug treatment selects for parasites carrying molecular markers of resistance to each drug. Maybe the authors can add "showed in Mali" in this sentence, else the citations (all of them previous publications of several authors of this manuscript) look like self-citations. 

Author Response

Comments and Suggestions for Authors -2

The authors describe here a study performed with 326 patients with uncomplicated falciparum malaria in Mali that measures the effect of antimalarial drug combination- artemether-lumefantrine, in terms of two genotypes - pfcrt and pfmdr1 genes that have been associated with drug resistance. The authors show that recurrent infections occur in 26% of the patients and these recurrent parasite population showed a significant shift towards the pfcrt K76 and pfmdr1 N86, pointing to association of these genotypes with AL drug treatment failure. The study is performed well and the results presented show statistical rigour. However, the authors do not show any novel findings and their results just confirm results from similar clinical studies in Africa, now extending it to Mali. This should not however exclude the work from publication.

Response: Thank you

Major comments-

1) Line 238- I could not understand this sentence. It would help if the authors could rephrase this in more simpler or layman terms.

Response: It is done. Thank you

2) Line 260 - At several instances like in this para, the authors talk about (resistant or susceptible), (mutant or wild), (parasites or alleles). It would make the discussion more readable and understandable if they could add the genotype notation (K76 or T76 etc) in brackets next to their assertions. I had to keep going back to their results or introduction to understand which allele is resistant/susceptible to which drug.

Response: genotype notation for example (K76 or T76 etc…) are added in brackets next to the assertions. Thank you

3) Line 261, Do you mean that allele selection occurs in reinfecting parasites rather than recrudescent parasites? Do you have enough data to support this assertion? I dont see any comparison of prevalence of mutations in reinfecting vs recrudescent. Given that there is only 8 cases of recrudescence in the dataset, probably we cant even make that comparison with reasonable statistical confidence. Maybe just say reason for recurrent infections is primarily reinfection and allele selection occurs in reinfecting parasites. 

Response: You are right, we haven’t statistical power about this data (N=8): “The reason for recurrent infections is primarily reinfection and allele selection occurs in reinfecting parasites”

4) Line 278 - You can be more specific as to how the pattern of pfmdr1 allele change is similar among all the studies.

Response: Okay, In the all these studies it was described a decreasing CQ exposure and increased lumefantrine exposure and this has selected the wildtype of Pfmdr1 and Pfcrt. Thank you

5) Line 284 - This paragraph need to be revised. I could not understand the point the authors are trying to make. 

Response: it’s done. Thank you

Minor comments-

1) Line 50 - remove apostrophe in Artemisinin.

Response: apostrophe in Artemisinin is removed, Thank you

2) Line 78 - I don’t know if "however" is appropriate here.

Response: We will change to “otherwise”, Thank you

3) Line 81 - "Here", a clinical study was performed...

Response: a clinical was performed… is added. Thank you

4) Line 82 - "pattern" to "patterns"

Response: pattern is changed to patterns, Thank you

5) Line 214 - "any death" to "no death"?

Response: any is changed to no, Thank you

6) Line 220 - missed bracket

Response: bracket is added, Thank you

7) Line 305 - "hypothesized" to "hypothesize"

Response: hypothesized is changed to hypothesize, Thank you

8) Line 63 - There are a ton of studies that have showed drug treatment selects for parasites carrying molecular markers of resistance to each drug. Maybe the authors can add "showed in Mali" in this sentence, else the citations (all of them previous publications of several authors of this manuscript) look like self-citations. 

Response:” We have started to describe the chloroquine resistance in the World/Africa before to discuss in the Mali”. Thank you